
# Variation and attribution of probable maximum precipitation of China using high-resolution dataset in a changing climate

Jinghua Xiong[1], Shenglian Guo[1*], Abhishek[2], Jiabo Yin[1], Chongyu Xu[3], Jun Wang[1], and Jing Guo[4]

[1]State Key Laboratory of Water Resources Engineering and Management, Wuhan University, Wuhan 430072, China.
5   [2]Department of Civil Engineering, Indian Institute of Technology Roorkee, Roorkee 247667, India
[3]Department of Geoscience, Oslo University, Norway.
[4]Power China Huadong Engineering Corporation Limited, Hangzhou 311122, China.

*Correspondence to*: Shenglian Guo (email: slguo@whu.edu.cn)

10   **Abstract.** Accurate assessment of the probable maximum precipitation (PMP) is crucial in assessing the resilience of high-risk water infrastructures, water resource management, and hydrological hazard mitigation. Conventionally, PMP is estimated based on a static climate assumption and is constrained by the insufficient spatial resolution of ground observations, thus neglecting the spatial heterogeneity and temporal variability of climate systems. Such assumptions are critical, especially for China, which is highly vulnerable to global warming in the premise of existing ~100,000 reservoirs. Here, we use the finest 15   spatiotemporal resolution (1d & 1km) precipitation dataset to present the spatial distribution of 1d PMP based on the improved Hershfield method. Current reservoir design values, a quasi-global satellite-based PMP database, and in-situ precipitation are used to benchmark against our results. The 35-year running trend from 1961-1995 to 1980-2014 is quantified and partitioned, followed by future projections using the Coupled Model Inter-comparison Project Phase 6 simulations under two scenarios. We find the national PMP generally decreases from Southeast to Northwest and is typically dominated by the high variability 20   of precipitation extremes in North China and high intensity in South China. Though consistent with previous project design values, our PMP calculations present underestimations by comparing with satellite and in-situ results due to differences in spatial scales and computation methods. Inter-annual variability, instead of the intensification of precipitation extremes, dominates the PMP running trends on a national scale. Climate change, mainly attributed to land-atmosphere coupling effects, leads to the widespread increase (>20%) of PMP across the country under the SSP126 scenario, which is projected to be higher 25   along with the intensification of $CO_2$ emission. Our observation- and modeling-based results can provide valuable implications for water managers under a changing climate.

## 1 Introduction

Over the past six decades, an increase in the frequency and intensity of extreme precipitation events have been documented in both observation- (Guerreiro et al., 2018; Martinez-Villalobos & Neelin, 2018; Visser et al., 2022; Zhao et al., 30   2023) and modeling-based studies globally (Donat et al., 2016; Kendon et al., 2017; Kunkel et al., 2013; Zhao et al., 2022).



This increase will relatively be more pronounced in the majority of the regions worldwide in a warming climate (Hirabayashi et al., 2013; IPCC, 2021; Kim et al., 2022), leading to the enhanced risk of the consequent floods and the associated multi-sectoral damage. Global damages due to floods amounted to an estimated $651 billion (USD) between 2000 and 2019 alone, which could increase by a factor of 20 by the year 2100 (Devitt et al., 2023; Winsemius et al., 2016). Traditional estimates of

such precipitation extremes and subsequent applications reliant on precipitation-sensitive information (e.g., flooding designs) have primarily relied upon the stationary climate assumption, which is inadequate for a large duration and in the warming climate (Visser et al., 2022). Another crucial application is Probable Maximum Precipitation (PMP), which is key to assessing the resilience of high-risk water infrastructures such as large dams and nuclear power plants, efficient water resource management, and impact assessment and strategic management of hydrological hazard adaptation and mitigation.

PMP, defined as the theoretical maximum precipitation for a given duration under modern meteorological conditions by the World Meteorological Organization (WMO), represents the upper limit of precipitation that is meteorologically possible over a watershed or a storm area of a given size at a certain time of a year (WMO, 2009). As an indicator of regional storm risks, PMP is physically dependent on various meteorological factors such as available atmospheric moisture content, moisture transportation efficiency, and persistent upward strength (Trenberth et al., 2003). In addition to the traditional flood frequency

analysis method, PMP also serves as the most severe condition to estimate the associated theoretical maximum flood for a certain project in the area (Hansen, 1987). Therefore, it plays a significant role in both the design of hydraulic structures (e.g., dams, reservoirs) and routing infrastructure and the assessment of regional weather hazards (e.g., storms) (Luo et al., 2018).

An underlying prevalent assumption of PMP estimation is the stationary climate leading to a static PMP value from observed meteorological data, such as wind speed, precipitation, and dewpoint, and maximized using empirical techniques,

meaning there is only a fixed PMP on a specific spatiotemporal scale (Visser et al., 2022; WMO, 2009). However, it has been significantly challenged when both observations and models show that the above key factors, i.e., wind and moisture, in forming PMP can change due to climate change and internal variability (Lalk, 2004; Mudd et al., 2014; de Winter et al., 2013; Gimeno et al., 2019; Richter and Xie, 2010; van Dilke et al., 2022). For example, the warming climate-induced increase in atmospheric moisture availability may favor the formation of extreme storm events (Liu et al., 2020). Besides, the natural

climate variability from annual to decadal scales (e.g., ENSO) may impact the accurate maximization of regional precipitation extremes, particularly with limited record length (Kenyon and Hegerl, 2010). A few previous studies have discussed the impacts of changing climate on PMP estimations over different regions of the world using global and/or regional climate models (Beauchamp et al., 2011; Rousseau et al., 2014; Rouhani and Leconte, 2016; Afrooz et al., 2015; Park et al., 2013; Lee and Kim, 2016; Visser et al., 2022). Specifically, Jakob et al. (2009) performed an early investigation in Australia and reported

increases in moisture availability in coastal regions that had tendencies to experience further projected increases under climate change. A global assessment from Kunkel et al. (2013) projected that future PMP values might intensify in the United States, contributed mainly by the higher levels of atmospheric moisture content, with a 20%~30% of increase in the United States by the end of the 21st century under a high gas emission scenario. Similar growth caused by a changing climate has been documented in India, Spain, and other parts globally (Sarkar and Maity, 2020; Monjo et al., 2023). However, opposite patterns



were also reported in a few regions, possibly due to the reduced actual moisture availability and wind speed by atmospheric dynamic constraints (Afzali-Gorouh et al., 2022; Yin et al., 2023). These inconsistent and contradictory findings imply the complicated mechanism and uncertainty in PMP estimations across regions and underscore the need for a holistic qualification of PMP considering non-stationary climate and at finer spatiotemporal scales.

Despite that the changeable PMP under a changing climate has attracted wide attention from hydrologists, most of the
previous studies primarily focus on the static scenario comparisons between history and the future (Jakob et al., 2009; Kunkel et al., 2013; Sarkar and Maity, 2020; Monjo et al., 2023; Afzali-Gorouh et al., 2022). Since the return periods corresponding to the PMP values outpace the longest return periods traditionally used in applied climatology products, major water retention and routing structures will likely experience the acute impact of climate change thus highlighting the elusive sense of security inferred from the assessments ignoring the climate-change-induced probabilities of extreme events (Kunkel et al., 2013).
Furthermore, the gradual transformation of the past climate and the partitioned contributions from various climate change sources also remain largely unexplored in the literature. Accounting for such realistic and crucial attributes and mechanisms is thereby necessary and topical, particularly for China, which has experienced persistent precipitation disasters over the past few decades (Gu et al., 2022). Covering a wide range of geophysical elevations and climate zones (Figures 1a and 1b), the country has faced increasingly significant spatial heterogeneities in extreme precipitation (Sun et al., 2017). It implies the
potentially intensified hydrological risk in different regions, which is more evident given the approximately 100,000 dams and reservoirs constructed until 2015, mainly for flood control (Figures 1c and 1d, MWR, 2016; Song et al., 2022). However, the systematic investigation of PMP in China was previously limited by inadequate spatiotemporal resolution and duration of precipitation measurements over the country and related climate modeling experiments.

Here, we use the precipitation dataset with potentially the finest spatiotemporal resolution (1d & 1km) covering 1961-
2014 to calculate the long-term average PMP distribution in China using the modified statistical method. The national estimations of PMP are benchmarked with a quasi-global PMP dataset based on satellite products and in-situ data from 2417 weather stations across the country (Figure 1c). The historical tendency in changing PMP is detected based on a 35-year running window method (consistent with the period of historical run of global climate models during 1980-2014), and the respective contribution from climate change and internal variability is partitioned. The role of land-atmosphere coupling, which
is an important contributor to climatic extremes, is further evaluated via an ensemble of global climate models. Finally, we project future changes in PMP in both the near and far future periods in both low-emission and high-emission scenarios relative to the baseline period (i.e., 1980-2014). All the results are separately discussed on different scales from river basins to country for efficient and effective policymaking inferences for the regional to national water managers.





**Figure 1: (a) The national map, major rivers, major river basin boundaries, and 1-km elevation of China. The digital elevation map is provided by A Big Earth Data Platform for Three Poles (Tang, 2019). The divisions of nine major river basins excluding a few coastal islands are provided by the Resource and Environment Science and Data Center of China (https://www.resdc.cn/), which include Haihe River basin (HRB), Yellow River basin (YRB), Huaihe River basin (HHB), Yangtze River basin (YTB), Southeast basin (including Taiwan Province, SEB), Pearl River basin (including Hainan Province, PRB), Northwest basin (NWB), Southwest basin (SWB), and Songhua and Liaohe River basin (SLB). The divisions of the 80 major secondary river basins outlined in green color are based on the regulations for the compilation of water resources protection planning of the Ministry of Water Resources (GIWRHPD et al., 2013). (b) The climate zones of China are produced by the China Meteorological Administration. The map is accessible on the Resource and Environment Science and Data Center of China (https://www.resdc.cn/), which is calculated using the national daily temperature and water measurements. The codes joined in the map represent the secondary climate zones, and more details can be found in previous references (Zhu, 1962; WCNR, 1959; Zhu, 1931). (c) The provincial administrative regions and locations of 2417 weather stations of China. The national map and provinces are made under the guidance by the standard map service of the Ministry of Natural Resources of the People's Republic of China (http://bzdt.ch.mnr.gov.cn/index.html). (d) The spatial distribution of 933 dams and reservoirs included in the Global Reservoir and Dam Database (GRanD) in China (Lehner et al., 2011).**





## 2 Materials and Methods

**2.1 High-resolution precipitation data**

A daily gridded precipitation data at a fine 1km spatial resolution covering the period 1951-2014 (namely the HRLT dataset) is used to estimate PMP over China (Qin et al., 2022). The HRLT precipitation data were interpolated using the best ensemble among various machine learning methods (i.e., boosted regression trees, random forests, neural network, multivariate adaptive regression splines, support vector machines, and generalized additional models; see Qin et al., 2022 for details) from

the 0.5°×0.5° observation-derived gridded precipitation from the China Meteorological Administration, combined with multiple external variables related to elevation, location, topography, and climate conditions (Zhao and Zhu, 2015). The superior spatial resolution of the HRLT dataset (i.e., 1 km) can prevent the effects of spatial heterogeneity in regional climate conditions on grid-scale PMP estimations. Apart from the major advantages of a longer period (1951-2019) and higher resolution, it has shown better accuracy than other widely used meteorological datasets in China like the China Meteorological

Administration Land Data Assimilation System (CLDAS, from 2017 to 2019 with ~7.5 km) version 2 and China Meteorological Forcing Dataset (CMFD, from 1979 to 2018 with ~12 km) (Shi et al., 2014; He et al., 2020). However, we selected the period 1951-2014 in this study to avoid the several unrealistic high precipitation values starting the year 2015 in the HRLT due to errors in the raw precipitation records, which could consequently lead to significant PMP overestimations (see Table S1 for details). Moreover, the locations and basic attributes (e.g., year of construction, year of decommissioning,

and storage capacity) of dams and reservoirs from China are collected from the GRanD dataset (Lehner et al., 2011) to analyze the temporal variations of total storage capacity of China. It is calculated as the ratio between the total storage capacity of dams within a certain region (e.g., river basin and the whole country) to the area, with the same unit as our PMP estimations (i.e., mm). Years of construction and decommissioning are considered in the computation. The GRanD dataset contains a total of 7320 dams worldwide based on the existing global lakes and wetlands database and national/continental statistics from

different sources, of which 933 are located in China (Figure 1d). All the records of the GRanD dataset are georeferenced and have undergone manual inspection and validation to avoid spatial inconsistency (between locations and attributes of dams) and redundancy. Since it only considers the dams with large sizes (>0.1 km$^3$), the number of included dams in China is much less than other similar collections (e.g., 97435 dams in CRD, see Song et al., 2022). However, the total storage capacity of dams in GRanD (670 km$^3$) accounts for ~70% of the CRD (980 km$^3$), the latter of which does not contain the necessary

attributes for temporal analysis (e.g., year of construction). By comparing the changes in PMP and the available storage capacity of dams with time, we can qualitatively measure the total capability of anthropogenetic efforts to store water from extreme precipitation. A higher difference between PMP and total dam storage capacity means more water cannot be stored in the basin reservoirs (needs to be consumed via evaporation and/or streamflow), and therefore, greater potential to translate to regional floods.



## 2.2 Validation of PMP estimations


Two independent data sources are collected to validate our 1d and 1 km PMP estimations using the HRLT dataset, including a quasi-global PMP dataset based on remote sensing products and a suite of national PMP results using in-situ precipitation records. The quasi-global PMP dataset is calculated based on the Integrated Multi-satellite Retrievals for GPM (Global Precipitation Measurement, namely GPMM hereafter) during 2000-2022 using the conventional Hershfield method

(Ekpetere et al., 2023). GPMM applies two existing corrections for the removal of the inversion problem caused by the relatively short period of IMERG product (i.e., 23 years) and for the correction of missing maximum precipitation samples. It has shown reasonable accuracy by comparing with NOAA ground gauges in Kansas, USA, from various time scales of 30 minutes to 24 hours (Ekpetere, 2021). Though sharing the same 1d timescale with the PMP estimations using HRLT, several key differences between GPMM and ours are worth mentioning. First, the GPMM is calculated using the classic Hershfield

algorithm combined with two statistical corrections above, which is different from our modified Hershfield algorithm (see details in Section 2.3). Second, the spatial scale of the GPMM is 0.1° (~11 km at the equator), which is much coarser than the HRLT dataset (1 km). Third, the period used for calculation in GPMM is 2000-2022, which is much shorter than our estimations that are based on HRLT data from 1961-2014. We additionally calculate the 1d PMP purely based on in-situ daily precipitation during 1961-2014 from 2417 weather stations of the country using the same modified Hershfield method (Figure

1c). The raw precipitation observations are provided by the China Meteorological Administration (https://www.cma.gov.cn/en2014/m/pc/) and the Resources and Environmental Science Data Center, Chinese Academy of Sciences (http://www.resdc.cn/, last access: 29 October 2023) upon research request. Despite the strict quality control (e.g., inspection of unphysical records) performed by the data providers, the spatial distribution of in-situ stations is uneven, with the number of available data decreasing from southeastern to northwestern parts of China, especially in the Qinghai-Tibetan

Plateau due to extreme natural environments to install and maintain the measuring stations. We use the bilinear interpolation method to extrapolate the PMP results based on the HRLT dataset to the locations of each grid cell of the GPMM and each field station of the precipitation network to facilitate inter-comparisons. The same procedure is repeated between GPMM and the in-situ precipitation results for better justification of our HRLT-based PMP estimations.

## 2.3 Statistical estimation of PMP


The methods of estimating PMP are generally classified into meteorological methods and statistical methods. The essence of the meteorological methods is the maximization of moisture factor and/or dynamic factor for a typical storm or an ideal storm model. However, it requires abundant hydro-meteorological data like dew point temperature and wind speed (Wang, 1999). The statistical approach is therefore recommended by WMO owing to its simplicity since it only needs precipitation data (WMO, 2009; Casas et al., 2008; Yang et al., 2018). It was originally developed by Hershfield (Hershfield, 1961) and

subsequently modified by Lin (1981) as:

$$PMP = (1 + K_m \cdot C_v) \cdot X'_n \qquad (1)$$



$$X_n' = \left(1 + \frac{3 \cdot C_v}{\sqrt{n}}\right) \cdot X_n \tag{2}$$

$$C_v = \frac{\sigma_n}{\bar{X}_n} \tag{3}$$

$$K_m = \frac{X_m - \bar{X}_{n-1}}{\sigma_{n-1}} \tag{4}$$

$$T_m = \frac{X_m - \bar{X}_n}{\sigma_n} \tag{5}$$

$$N_m = T_m^2 + 2 \tag{6}$$

where Eqs. (1) – (4) represent the generalized formula of Hershfield's algorithm (Hershfield, 1961), which is based on the product of mean annual maximum precipitation and the maximization factor $K_m$. $X_m$ is the annual maximum precipitation series, and $\sigma_n (\bar{X}_n)$ is its standard deviation (mean) value, with $\sigma_{n-1}(\bar{X}_{n-1})$ meaning the same as the series but excluding the maximum value. Lin (1981) revised the expression of $X_n$ to correct the sampling error in averaging annual maximum precipitation ($X_n'$). $C_v$ is the coefficient of variation of annual maximum precipitation series. An additional constraint is given to the ultimate PMP estimations in Eqs. (5) – (6) to determine if the length of the precipitation series has satisfied the requirement to capture the inter-annual variability of precipitation extremes, serving as quality checks of PMP results. We perform all the PMP calculations for each 1 km grid of China, which can reasonably be considered as a hydro-meteorological homogeneous region to capture consistent characteristics of precipitation extremes. The ultimate PMP estimates are additionally multiplied by 1.13 to reflect the influences of a single fixed precipitation record frequency on yielding true maxima (WMO, 2009). Above computations are performed for each 1 km grid cell over the country (~1,400,000) during each considering period, generating a comprehensively high-resolution and time-varying detection of national PMP (see details in the next section).

## 2.4 Detection and partition of PMP trends

Given the fact that the changing climate may influence the PMP estimates of a specific region over a specific period, we separately compute the PMP of each grid during different 35-year running windows (i.e., 1961-1995, 1962-1996, …, 1980-2014). It is selected to be consistent with the period of the historical run of global climate models during 1980-2014 (refer to Section 2.5 for details). We consequently obtain a total of 20 subsets of PMP estimations for each 35-year period during 1961-2014, which are subsequently used to calculate trend slopes using the linear regression method, with the significance level identified based on Mann Kendall's Z-statistics (5% in our study) (Xiong et al., 2020; Mann, 1945; Yin et al., 2021). Furthermore, looking back at Eq. (1), we reformulate the formation of PMP as two key factors of intensity and inter-annual variability of extreme precipitation and write it as.

$$PMP = K \cdot X_n' \tag{7}$$





where $K$ is the integrated maximization factor equivalent to the item $(1 + K_m \cdot C_v)$ in Eq. (1). We consider $X'_n$ to reflect the intensity of extreme precipitation events since they are closely related to the available atmospheric moisture and persistent upward motion that are sensitive to atmospheric warming (Loriaux et al., 2016). The $K$ factor is an indicator of inter-annual variability of precipitation extremes during a certain period as it is derived from the standard deviation and maximum value of annual maximum precipitation (standardized by the long-term mean). Eq. (7) can further be transformed into a logarithmic

form:

$$lg_{PMP} = lg_K + lg_{X'_n} \qquad (8)$$

In such case, a multiple regression model between these logarithmic items can be constructed to quantify the respective contributions from intensity ($X'_n$) and variability ($K$) factor, where the trend of $lg_{PMP}$ can be sourced from constituent $lg_{X'_n}$ and $lg_K$. Their relative contribution rates (%) of trends can thereby be estimated as $\frac{s_{lg_K}}{s_{lg_{PMP}}}$ and $\frac{s_{lg_{X'_n}}}{s_{lg_{PMP}}}$, respectively ($S$ is the

trend slope). Note all the actual trend slopes are calculated using the original variables, while the logarithmic transformation is only applied to calculate the relative contribution rates of both $X'_n$ and $K$ factor.

Furthermore, as a major contributor to precipitation extremes, land-atmosphere coupling effects have received special attention by comparing ensemble global climate model (GCM) simulations from the historical simulations of Coupled Model Inter-comparison Project phase 6 (CMIP6) and the Land Surface, Snow and Soil Moisture Model Inter-comparison Project

(LS3MIP) during 1980-2014 (a time slice of the observational 35-year running results 1961-1995, 1962-1996, …, 1980-2014) (van den Hurk et al., 2016). Their only difference lies in the prescription of dynamic land states of the LS3MIP (namely LFMIP-pdLC experiment), including snow and soil moisture based on the long-term climatology during 1980-2014. This experiment does not consider the seasonal variability of soil moisture, thus diminishing the influences of land's feedback on the atmosphere, providing a good way to remove the land-atmosphere coupling. The GCMs we selected include CMCC-ESM2,

CNRM-CM6-1, EC-Earth3, IPSL-CM6A-LR, MIROC6, and MPI-ESM1-2-LR models, which are the only models that provide the daily precipitation in both CMIP and LFMIP-pdLC experiments currently. However, we note a few models do not provide specific flux variables (e.g., latent heat flux) that can be used to further explain the potential mechanisms of land-atmosphere coupling to influence PMP, which are also included in our analysis to extend the data availability and reduce the uncertainty of a single model (Table S2).

**2.5 Projection of PMP under climate change**

Using the daily precipitation data of the same GCMs from the Scenario Model Inter-comparison Project (SMIP) and LFMIP-pdLC as those in the historical CMIP experiments, we project the temporal variations of PMP during 1980-2099 under the Shared Socioeconomic Pathways 1-2.6 (SSP126) and 5-8.5 (SSP585) scenarios, which represents the least and most extreme pathway with high greenhouse gas emissions (2.6 and 8.5 W/m² of forcing in the year 2100), respectively, together

with the slow and rapid social-economic growth (O'Neil et al., 2016; Eyring et al., 2016; Pörtner et al., 2022). Comparisons





between future and historical periods for the two most extreme scenarios allow the understanding of the bounding influences of climate change on future PMP conditions. Specifically, we quantify the percentage changes in PMP between the middle and end of the 21st century (2030-2064 and 2065-2099, respectively) and the reference historical baseline (1980-2014) using the same models from the CMIP and SMIP projects, which represent the predicted PMP changes in the near and far future.

Moreover, we conduct the inter-comparison between SMIP and LFMIP experiments to examine the potential influences of land-atmosphere coupling effects on PMP shifts under climate change. The deviations across models are additionally illustrated in the supplementary files to reflect the model uncertainty (Zhang and Chen, 2021; Jia et al., 2023).

## 3 Results

### 3.1 Spatial distribution of PMP

Long-term average 1d PMP and its constituting factors ($X'_n$ and $K$) during 1961-2014 are estimated over China to reveal their spatial patterns (Figure 2). We observe a general three-step spatial distribution with $X'_n$ generally decreasing from southeast to northwest, especially high for the coastal regions of Hainan and Taiwan islands (refer to Figure 1c for their locations) (>120 mm/d locally). High values are also discovered in mountainous areas like the southern Himalayan region and the middle and lower reaches of the Yangtze River basin (Figures 1a and 2a). However, the regional $X'_n$ keeps below 15 mm/d

over the majority of northwestern China due to the arid climate (Figures 1b and 2a). Contrary to the variable $X'_n$ representing the intensification of precipitation extremes, the $K$ factor captures a gradually decreasing tendency from northwestern to southeastern China, ranging from 17.2 to 1.2. It indicates that the inter-annual variability of precipitation is stronger in arid northwestern China than in the humid regions in the southeastern parts. A few regions with significant variability are discovered in North China, the southern part of the Qinghai-Tibetan Plateau, and scattered regions of South China (Figure 2b),

which are possibly related to the local geophysical and climatic conditions (e.g., elevated terrain and coastal storm). Consequently, the contributions of divergent spatial patterns in $X'_n$ and $K$ leads to the complex distribution of PMP over China. It is characterized by the overall 'high in southeastern and low in northwestern' distribution similar to $X'_n$, with a few regions highlighted by overwhelming PMP strengthened by local $K$ factor (e.g., Huaihe and Haihe River basins of North China) (Figure 2c). Central Yangtze River basin, where both factors forming PMP ($X'_n$ and $K$) present relatively high values, is highlighted

by the large amplitude of PMP. Specifically, the area-averaged values for the Yangtze, Southeast, and Pearl River basins are 131, 225, and 196 mm/d, respectively (109 mm/d for the whole of China). Overall, it coincides with the national dam and reservoir distribution to imply the regional flood potential and consequential human interventions to alleviate such impacts (Figures 2c and 1d). The negative linear regression between the upstream drainage area and PMP of 52 major water conservancy projects of China ($R^2$=0.53, $p$<0.05) is reasonably reconstructed from our PMP results from over 80 major

secondary river basins ($R^2$=0.39, $p$<0.05) (Table S3 and Figure 2d). Differences in the slopes are mainly induced by varying spatiotemporal scales for calculations and equip us with improved insights into the scale dependencies of the estimated PMP.



Comparisons in the spatial distribution of PMP with previous estimates demonstrate the robustness of our HRLT-based results. Our estimations of $X'_n$ well reproduce the national distribution of historical records of daily precipitation maxima, except for the Inner Mongolia Province, where a historical precipitation extreme of ~1400 mm/d happened in 1977 was reported (Figures 1c, 2a, and 3b). However, scale differences between ground stations and grid cells lead to neglecting such events in PMP calculations, which should deserve more attention for regional investigations. Furthermore, the spatial distribution of our PMP results corresponds well with a previous preliminary estimation of the national PMP map based on in-situ data, which has been transformed from the original contour line to a gridded rendering map for better visualization (Figure 3a). The coherent high PMP is not only located in East China along the coastline but also in a few arid regions in northwestern China, as well as the southernmost part of Xizang Province (Figure 1c). Independent comparisons with two suites of PMP estimations over China additionally suggest that our HRLT-based PMP is able to illustrate a similar spatial distribution to that of in-situ results where abundant ground precipitation is available (e.g., East and South China) (Figure 3c). More importantly, it depicts the PMP distribution for data-scarce regions like Xinjiang and Tibet Provinces in Western China, where very limited information can be extracted from in-situ results (Figure 1c), which is supported by the GPMM results that is derived from remote sensing precipitation product (e.g., relatively high PMP in South Tibet) (Figure 3d). However, the GPMM data presents obvious overestimations of PMP for nearly the whole of the country, reaching ~4300 mm/d by comparing with previous investigations and the in-situ results (Figures 3a-3c), which is related to the systematical overestimation of GPM IMERG product over China (Pan et al., 2023). Such overestimation can propagate into the calculation of $K$ factor and, therefore, further unrealistically amplify the PMP. The differences in the computation methods with GPMM and the relatively short period (i.e., 2000-2022) may also contribute to the overestimated PMP. Moreover, more specific regional distributions of PMP, e.g., the high PMP values in the south-north Taihang Mountains in North China, are highlighted by the HRLT-based PMP. While this is not seen in the GPMM because HRLT-based PMP was calculated on a much finer spatial resolution (1 km) than GPMM (0.1°, ~11km).

Quantitative validation is performed on various scales among PMP estimations from HRLT, GPMM, and in-situ results (Figure 4). Relatively good correlations between PMP estimations from HRLT and the other two subsets are found on the grid scale, with the Pearson Correlation Coefficient (CC) of 0.65 and 0.66 to GPMM and in-situ results, respectively (Figures 4a and 4b). However, the significant overestimation of GPMM is reported by comparing it to HRLT results, where a line with a slope of 2.51 is fitted, consistent with the overall estimation of spatial distributions (Figures 2c and 3d). This slope is apparently higher than that between in-situ results and HRLT data (1.54), indicating the effectiveness of our HRLT results. We also report similar overestimations of GPMM to the in-situ results and decreased correlations (CC=0.52) (Figure 4c). Examination results on the region scale also reveal a similar situation, with better agreement between HRLT and in-situ results (CC=0.96) than that with GPMM due to its significant overestimations at a regression slope of 3.67 (Figures 4d-4e). The regional estimate of PMP from GPMM is nearly double the in-situ results over different river basins, fitting a line of 0.55 between both subsets (Figure 4f). Therefore, the HRLT-based PMP shows relatively better accuracy than the GPMM dataset in China by comparing with the in-situ-based results, though it also presents moderate overestimations than in-situ data.

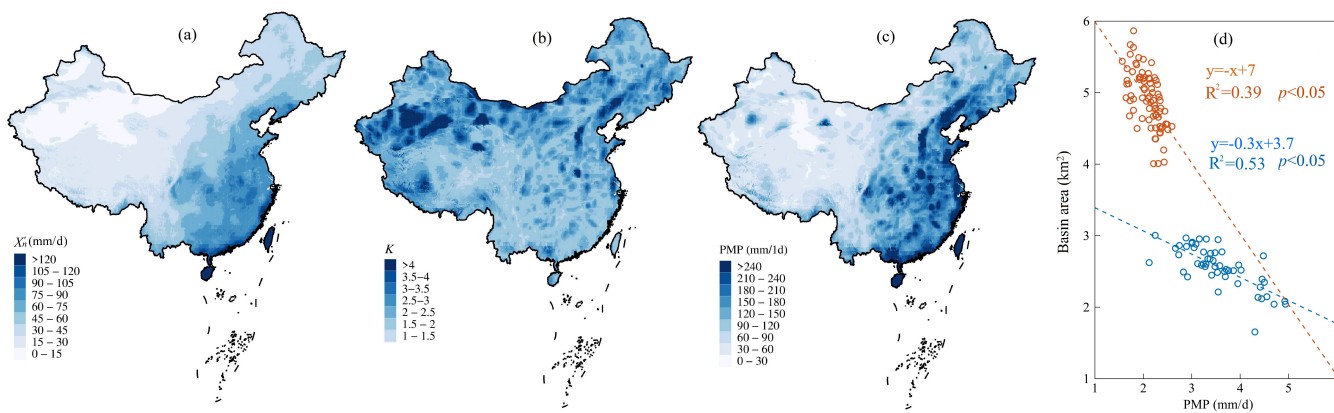

**Figure 2: Spatial distribution of (a) $X'_n$, (b) $K$ factor, and (c) PMP based on HRLT dataset during 1961-2014. The grid cells and stations where the minimum length of years to calculate PMP is not satisfied are masked out for clarification. (d) Scatter plots between PMP estimations and catchment area of 52 major Chinese water conservancy projects (blue) (GIWCHPD, 1982, 1990, Table S3) and 80 secondary river basins (orange). Both PMP estimations and catchment areas have undergone logarithmic transformations for better visualization.**

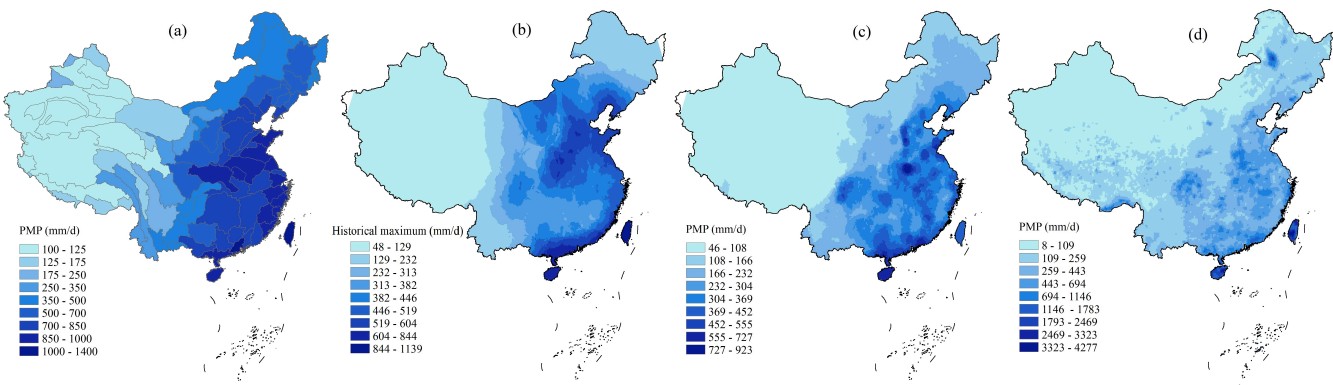

**Figure 3: (a) Spatial distribution of field-based PMP over 80 secondary river basins (Wang, 2002). (b) Spatial distribution of recorded historical maximum daily precipitation (Wang, 2002, Table S4). (c) Spatial distribution of PMP based on in-situ daily precipitation during 1961-2014. (d) Spatial distribution of PMP results from GPMM database.**





**Figure 4:** Scatter plots between PMP estimations from (a, d) HRLT and GPMM, (b, e) HRLT and in-situ precipitation, and (c, f) GPMM and in-situ precipitation dataset on (a, b, c) the grid/station scale and (d, e, f) the basin scale. The dashed gray and solid black lines represent the 1:1 line and fitted linear regression line, respectively. Sub-figures (a-c) are heatmaps where high (low) point density is translated to yellow (blue) colors. CC means the Pearson Correlation Coefficient. Black dots and red triangle in sub-figures (d-f) represent the different river basins and the whole country, respectively.

## 3.2 Variations and attributions of PMP

Firstly, based on the assumption of a static climate, the spatial distribution of PMP over China is evaluated using the high-resolution HRLT dataset and validated with in-situ results and GPMM data. However, since the changing climate is a widely acknowledged fact by the community, significantly challenging the accurate estimation of PMP (Piao et al., 2010). Secondly, we estimate the changes of PMP and its constituting factors (i.e., $X'_n$ and $K$) over different time slices of 35 years from 1961-1995 to 1980-2014 to detect such influences (Figure 5). On a national scale, we observe a reduction of $K$ from the period 1961-1995 (2.3) to 1971-2005 (2.23), followed by an increase until the period 1977-2011, and nearly stable at around 2.28 in the following years (Figure 5a). Contrary to $K$, $X'_n$ presents a stably increasing tendency over all the periods, meaning a relative increase of 3% if compared to the first period, i.e., from 43.5 (1961-1995) to 44.6 mm/d (1980-2014) (Figure 5b). Consequently, the national PMP shows a pattern that is dominated by the $K$ factor, including a minor decline before the period





1971-2005 and a continuous increase afterward. The accelerated rise of PMP from 1977-2011 should be highlighted, which results from the joint contribution of the increase in $X'_n$ and $K$ factor (Figure 5c). It can be the result of the intensification of both climate variability (e.g., El Niño–Southern Oscillation events) (Huang and Stevenson, 2023) and anthropogenic forcings (e.g., irrigation and urbanization) (Wu et al., 2021; Han et al., 2022). Overall, the national average PMP increased from 106.5 to 109.5 mm/d between the period 1961-1995 to 1980-2014, equivalent to a 3% increase with baseline from the first period. This growth also coincides with the steady rise of the total storage capacity of the dams and reservoirs, implying the artificial efforts to alleviate the impacts of increasing and more intense precipitation extremes. On the other hand, it also provides advance warning to the water resource managers that more constructions are needed in the future in case of the overwhelming increase rate of PMP than reservoir and dam constructions, even though the total reservoir capacity of the country has increased by ~80% from 1961 (35.9 mm) to 2014 (64 mm) (Figure 5d).

The inter-annual trend of 1d PMP during the 35-year running window from 1961-1995 to 1980-2014 is firstly estimated for each grid cell (Figure 6a). The spatial distribution of national PMP trend is featured by the widespread increase of PMP across North China with regional hotspots in Inner Mongolia and Heilongjiang Provinces (>5 mm/d/a) (refer to Figure 1 for their location). The region-averaged result for the Yellow and Songliao River basins where they are distributed is 0.41 and 0.14 mm/d/a, respectively. Another region with a significant PMP increase is in the Southern part of the country, comprising mainly the central Yangtze and Pearl River basins (0.7 and 0.27 mm/d/a, respectively), where slopes are higher than 7.5 mm/d/a locally. In addition, both significant ($p<0.05$) increasing and decreasing trends are detected in the scattered regions of Northwest and East China. Such a distribution provides a piece of evidence on the necessity of incorporating the non-stationarity of a climate system in the calculation of PMP as well as the pressing need to consider its long-term change behaviours.

The drivers of the PMP trend are attributed to its two contributors (i.e., $X'_n$ and $K$) according to Eq. 8. It is found the national distribution of PMP is mainly controlled by the latter in the spatial domain (Figures 6b to 6e). The relative contribution of the trend in the $K$ factor accounts for higher than 100% of both increasing and decreasing trends of regional PMP over most of the country and rises up to 400% for certain areas in Northern and Western China (Figure 6e). Another variable, the $X'_n$, presents a divergent pattern in the remaining parts of the country, with contribution rates lower than 50% (Figure 6d). However, apart from the consistent growth of the $X'_n$ in Southeast China (trend rates: 0.25 to 0.75 mm/d/a), there are significant increases over the Northwest part, though the change slopes are generally lower than 0.25 mm/d/a (Figure 2b). Differently, the $K$ factor mainly illustrates growth in Northern China and the neighbouring Qinghai-Tibetan Plateau, even though a few regional hotspots with rapid decline are found in the Yangtze and Pearl River basins (Figure 6c). These findings underpin our hypothesis that a static climate assumption to calculate PMP is not appropriate for most areas of China due to significant increasing/decreasing trends, which are overall caused by the changes in the inter-annual variability of precipitation extremes instead of its intensity, though the latter has demonstrated widespread increase over most of the country. On the national scale, the PMP increases at a rate of 0.08 mm/d/a, of which 71% (29%) is caused by the increasing $K$ factor ($X'_n$). It is also the





governing factor in most river basins of the country, where the highest contribution of 98% is in the Pearl River basin (Figure 6f), with the opposite pattern observed in the Haihe River basin, where the $X'_n$ contributes most, up to 38%.

**Figure 5**: Temporal changes of 35-year estimate of (a) K factor, (b) $X'_n$, (c) PMP, and (d) total reservoir capacity from 1961-1995 to
1980-2014 in China. The x-axis label 1995, 1996, …, 2014 means the period 1961-1995, 1962-1996, …, 1980-2014, respectively.





**Figure 6**: Trend slopes of (a) PMP, (b) $X'_n$, and (c) K on the daily scale of the moving 35-year periods from 1961-1995, 1962-1996, …, to 1980-2014 over China. Contribution of (d) $X'_n$ and (e) K factor to the changing 1d PMP. The grid cells whose trend values do not reach a 0.05 significance level are masked out. (f) Scaled contribution of different variables to the changing PMP in different river basins. Please refer to Figure 1 for details of the regional abbreviations.

### 3.3 Response of PMP to the changing climate

A static period from 1980-2014 is chosen to evaluate the prediction capability of ensemble GCMs. By comparing the historical PMP results from the CMIP experiment with the HRLT PMP results during the same period, we observe coherent distributions among them in terms of $X'_n$, K factor, and PMP (Figure 7). However, differences in the amplitude of these variables exist due to divergent spatial scales (1 km vs 1°) upon PMP calculation, causing the larger cells (1°) generally possess lower values, with more local details found in the former (1 km). No significant differences are observed between CMIP and LFMIP-pdLC experiments, meaning the subtle effects of land-atmosphere coupling in the past. Individual simulations from



single models are presented to analyse the inter-member uncertainty (Figure S1). We find the ensemble median PMP is the balanced result of overestimated values from CMCC-ESM2 and MIROC6 and the underestimated values from MPI-ESM1-2-
LR, which is caused by the requisite interpolation from the native coarse model resolution (~2°). Overall, the consistent PMP distributions in the ensemble median of models and observational results indicate the effectiveness of GCM predictions, which are, therefore, further applied to project future changes under climate warming.

We quantify the relative changes in 1d PMP between future periods and baseline (1980-2014) under two climate change scenarios (Figure 8). A widespread increase during the near future (2030-2064) is projected across nearly the whole country
from the SMIP experiment under the SSP126 scenario, particularly in the Southern coastal region, Northeast China, Central part of the Yangtze River basin, West of Inner Mongolia, and the Yarlung Zangbo River basin located in Southwest China (Figure 8a). The percentage change generally exceeds 20% and reaches up to 60% for certain regions, which results from the intensification of both $X'_n$ and $K$ factor (Figures. S2a and S4a). Specifically, the overall increasing PMP is mainly caused by the growth of $X'_n$ at the national domain, with the intensification of $K$ factor over specific regions. However, such an increase
is significantly dampened (and even reversed) in the LFMIP-pdLC experiment due to the widespread reduction of $X'_n$ except for a few regions around the western and northern boundaries of China (Figures 8c, S2c, and S4c), with the $K$ factor almost unchanged during the same period (Figure S4). The overestimated PMP results between SMIP and LFMIP-pdLC are mainly located in the Southern tropics and arid and semi-arid zones of Northwest and Northeast China, which are caused by the underestimated $X'_n$ in the LFMIP-pdLC with $K$ factor slightly reduced (Figures 8a and 8c). However, the increase in PMP of
the scattered regions in the Northwest China persistently exists in both experiments. No significant differences between near (2030-2064) and far future (2065-2099) projections are discovered in both SMIP and LFMIP-pdLC experiments (Figures 8a-8d). To conclude, the projected PMP increase reaches 20% and 17% for the whole country during the near and far future periods, respectively, according to the SMIP experiment, of which the Southwest (31%) and Southeast (21%) basins are, correspondingly, the highest. The percentage changes are reduced to only 2% (near future) and 0% (far future) for the LFMIP-
pdLC experiment. Furthermore, we observe the continuous intensifications of PMP in the SSP585 scenario compared to the SSP126 scenario in the SMIP experiment, with the overall decrease of PMP in the LFMIP-pdLC experiment (Figures 9, S3, and S5). These changes are caused by the increase/decrease in the $X'_n$ from the SSP126 to SSP585 scenario during the SMIP/LFMIP-pdLC experiments, with the $K$ factor almost unchanged among scenarios. The PMP increases to 51% and 43% for the SSP585 scenario from the SMIP experiment during the near and far future compared to the baseline period, much
higher than the LFMIP-pdLC results (-1% and -5% for the near and far future) (Figure 9e). These findings suggest that the land-atmosphere coupling controls the increase in PMP for the majority of China mainly by influencing the intensity of precipitation extremes (i.e., $X'_n$). However, the climatic change unrelated to the land-atmosphere coupling governs the strengthened PMP in Northwest China, where significant increases in PMP are detected due to the growing variability of precipitation extremes (i.e., $K$ factor). They imply the compound risk of increasing intensity and variability of precipitation
extremes under climate change. These findings are consistent with previous global assessments using the GLACE-CMIP5 framework, which found a decrease in the annual sum of daily precipitation (>95 percentile) after removing soil moisture





variability (i.e., representative of land-atmosphere coupling) in South China (Lorenz et al., 2016). However, it also indicated enhanced variability of heavy precipitation in water-limited regions due to increased latent heat flux that tends to increase evaporation and precipitation (Berg et al., 2014). It is different from our examinations over the semi-arid and arid zones across the country (Figure 1b), possibly due to the divergent response of latent/sensible heat flux to atmosphere states spatially (Wu et al., 2023).

**Figure 7**: **Estimates of 1d (a, d) $X'_n$, (b, e) $K$ factor, and (c, f) PMP from the (upper panel) HRLT and (lower panel) ensemble mean of the CMIP experiment during 1980-2014 over China.**



**Figure 8**: **Multi-model mean percentage changes in 1d PMP from (a, c) 2030-2064 and (b, d) 2065-2099 period to 1980-2014 under SSP126 scenario over China from (a, b) SMIP and (c, d) LFMIP-pdLC ensembles. (e) Regional summary of the percentage PMP changes. Please refer to Figure 1 for more details on the regional abbreviations.**








**Figure 9**: **Same as Figure 8, but for the SSP585 scenario.**



## 4 Discussions

### 4.1 Comparisons with previous studies

Quantitative assessment with design values of large hydropower projects and in-situ estimations of PMP has presented
a contradictory conclusion, i.e., overestimation of the water projects (Figure 2f) and underestimation of in-situ results and the
GPMM database (Figure 4). This fact suggests more justification should be carried out by comparing with previous research.
A few regional studies have calculated 1d PMP over different parts of China (Svensson & Rakhecha, 1998; Yang et al., 2018;
Zhou et al., 2020). For example, Svensson & Rakhecha (1998) used the moisture maximization factor to estimate PMP over
the Hongru River basin of the Huaihe River basin in eastern China, resulting in a result of 460 mm/d that is generally within
the range of our 1km PMP map of the corresponding area (200~600 mm/d, Figure 2c). Zhou et al. (2020) applied the storm
transposition method to estimate PMP for a small ungauged catchment in northern China from 118°20'E~118°26'E and
40°26'N~40°30'N. The results change from 397 to 570 mm/d at most stations, with an extreme value of 1026 mm/d in
Zhangmu, Hebei Province (see Figure 1c). They are overall higher than our gridded estimations using HRLT, with PMP
approximately fluctuating between 80 and 200 mm/d. Such difference may arise from the distinctive calculation methods (i.e.,
hydrometeorological method vs statistical method) and data length with our study, as most historical maximum precipitation
occurred prior to the beginning year of the HRLT dataset (1961). It suggests the sensitivity of PMP estimations on different
computation methods and data representativeness for valid precipitation extremes. Another example in western China is the
calculation of PMP for the Nujiang River basin (part of the Southwest basin, Figure 1a) (Liu et al., 2016). The study uses a
gridded precipitation dataset to estimate PMP based on the model storm amplification approach. It discovered the PMP
increases from upstream to downstream within the basin, and the value changes from 15.4 to 99.7 mm/d. The spatial
distribution (Figure 2c) and amplitude (28.7~87.8 mm/d) are quite similar to ours. Using three remote sensing precipitation
products and the statistical method, Yang et al. (2018) discussed the potential of satellite data to estimate PMP in poorly gauged
regions by taking the Dadu River basin in western China (located in the upstream Yangtze River basin, Figure 1a) as an
example. It pointed out the overestimation of multiple satellite products and recommended the PMP of 52~519 mm/d over the
region, which is relatively higher than our result of about 29~279 mm/d. The discrepancies again imply the overestimated
PMP from the HRLT is similar to our comparisons with national field precipitation results (Figure 4). However, a consistent
spatial variability and distribution is reported between Yang et al. (2018) and our study (Figure 3a), where PMP generally
increases from upper to lower reach. We also find overestimations of HRLT PMP (~350 mm/d) in the Hong Kong Island of
South China, which is apparently lower than results based on site data (e.g., 1753 mm/d in Lan et al., 2017 and Liao et al.,
2020). Such underestimations, on the one hand, are the consequence of different calculation algorithms, data sources, and
uncertainties. On the other hand, they reflect the differences in spatial scales between field and grid cell PMP estimations.
Previous studies generally take the highest estimation among various weather stations in a region as the final PMP, while the
HRLT highlights the average PMP for each high-resolution 1km grid cell. Indeed, our approach tends to follow the definition
of PMP more strictly, i.e., the theoretical maximum precipitation for a given duration under modern meteorological conditions,



which should happen on an area scale instead of a point domain (WMO, 2009). This scale difference is further highlighted in a global study that quantified the change of 1d PMP and mean annual maximum daily precipitation (AMDP) using a 0.5° resolution global precipitation dataset (Sarkar & Maity, 2021). The mean AMDP of grid cells over tropical zones with high precipitation and low seasonality (e.g., Southeast Asia near Hong Kong and Taiwan Islands) generally range from 50 to 150 mm/d (see Figure 5 of Sarkar & Maity (2021)), much lower than our HRLT estimates (Figure 2a) and previous station-based

estimates (e.g., Table 5 of Lan et al., 2017) due to larger grid cells for computation (~50 km). Moreover, it indicated a significant increase of PMP in the southern and northeastern parts of China by comparing the PMP results of two periods (i.e., 1948-1977 and 1979-2012), which coincide with our spatial distribution of PMP trends (Figure 6a), even though over different calculation periods.

Despite some differences between previous regional investigations that are derived from divergent datasets, methods,
and spatial scales, the first high-resolution (1 km) PMP map generated over China captures the spatial distribution at a local scale very well. Moreover, all the previously mentioned studies estimate PMP under the assumption of a static climate and neglect the variability of climate systems. This point is addressed in this study by separately calculating the PMP for each moving 35-year time period, along with an attribution framework proposed to track the sources of PMP changes. Anthropogenetic climate change, which is not adequately discussed in previous studies, is also investigated using an ensemble
of global climate models under different scenarios and periods. In a nutshell, this study constructs the first national high-resolution PMP map and quantitatively detects the changing climate influences on PMP estimations in the past and future.

## 4.2 Potential pathways of land-atmosphere coupling to PMP

Linkages between land-atmosphere coupling and climate extremes have received much attention from the community over the years by means of observations and models (e.g., Koster et al., 2004; Zhou et al., 2019). However, most previous
analysis focus on the mechanisms of land-atmosphere coupling to induce the hot extremes in the near-surface interface, leaving the rationales behind the extreme precipitation events (and PMP) still poorly understood (Lorenz et al., 2016). Nevertheless, a basic consensus is that land surface states (typically soil moisture) alter the atmospheric processes by modulating the allocation of sensible and latent heat flux of the energy budget, including a positive and a negative way (Seneviratne et al., 2010). On the one hand, the increasing wetness of soil can provide more available moisture to be evaporated into the near-surface atmosphere,
leading to higher evaporation (or upward latent heat flux); on the other hand, the increased evaporation can inversely reduce the available soil moisture. While this reduction should be lower than the increased soil moisture to maintain the interactions between soil and precipitation; otherwise, the soil would become drier. In this case, the elevated net evaporation can further influence the precipitation, by enhancing the moisture supply for the planetary boundary layer (PBL) to affect the atmospheric heating rates and cloud formation (Zheng et al., 2015). However, such 'second-hand' influences are complex due to the
multiple dynamic and thermal processes involved. Both positive and negative correlations have been reported from the previous modeling outputs and observed results depending on different regions and seasons (Diro et al., 2014; Wu et al., 2023).




Based on prior knowledge of the mechanisms of land-atmosphere coupling, we detect the percentage changes of the annual daily maximum of heat fluxes of future scenarios compared to the baseline (Figures 10 and 11). We discover the national increase in the annual daily maxima of latent hear flux that is most obvious in western parts of China according to the

SMIP experiment, which is spatially consistent with the increase in $X'_n$ (Figures 10 and S2). Such increase disappears in the LFMIP-pdLC experiment and keeps a similar spatial pattern to the $X'_n$, i.e., the decrease in the majority of the country with the regional increase in the West and South. Furthermore, these reported changes in the latent heat flux show no apparent deviation between the near and far future; however, they show significant positive sensitivity along with the enhancement of the gas emission scenarios. In addition, we also find the corresponding variations in the sensible heat flux that are opposite to the latent

heat flux, with the exception of Northeast China, where both fluxes increase in the LFMIP-pdLC experiment (Figures 10 and 11). Moreover, the strengthened changes in the sensible heat flux are observed in both ensembles. Based on the above analysis, it can be inferred that the land-atmosphere coupling can enhance the intensity of precipitation extremes by increasing the supply of latent hear flux (i.e., evapotranspiration) at the expense of reduced sensible heat flux and such impacts can be relatively stronger under a higher gas-emission-scenario.

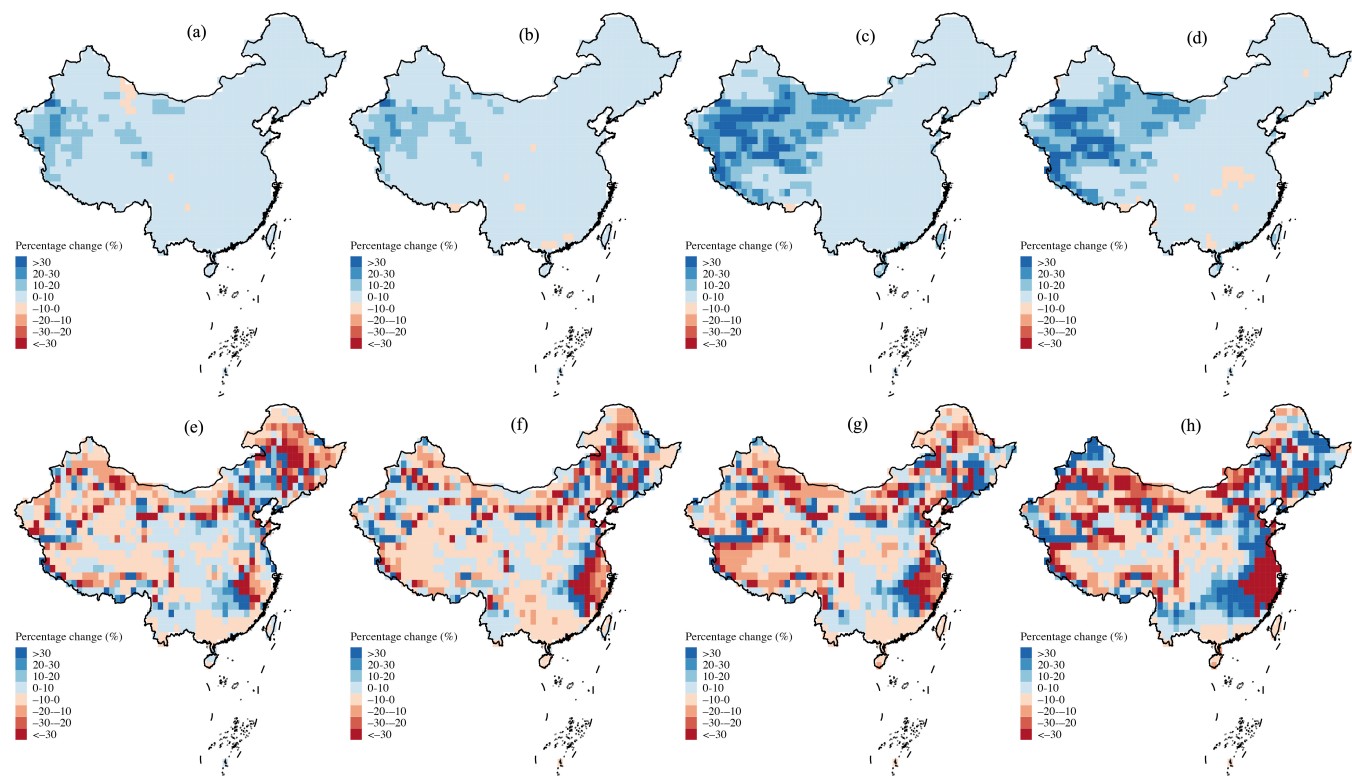


**Figure 10**: Spatial distribution of the multi-model mean percentage changes in latent heat flux from (a, c, e, f) 2030-2064 and (b, d, f, h) 2065-2099 period to 1980-2014 under (left two columns) SSP126 and (right two columns) SSP585 scenarios over China from (upper row) SMIP and (lower row) LFMIP-pdLC ensembles.



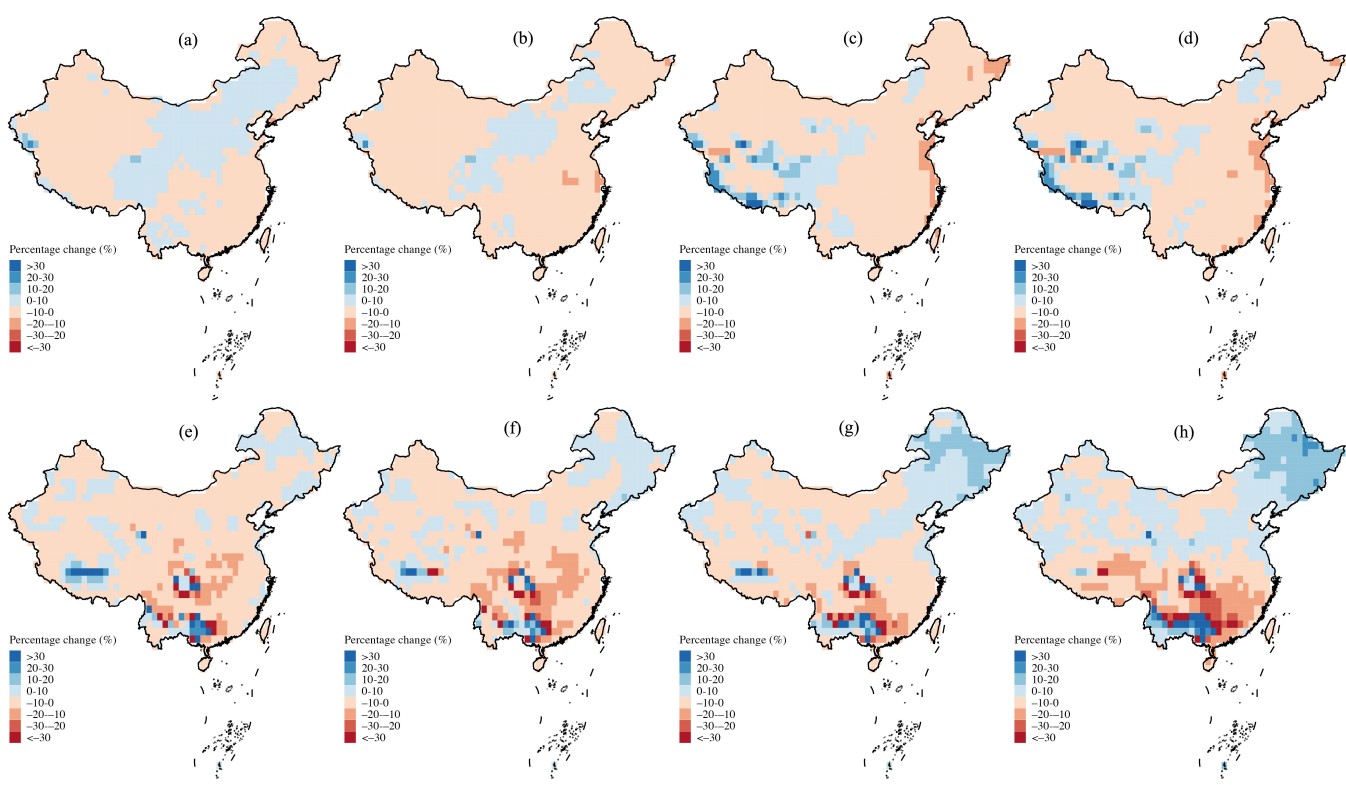

**Figure 11**: **Same as Figure 10, but for the sensible heat flux.**

### 4.3 Limitations and future outlooks

Although this study has provided the first national high-resolution PMP map and the quantitative evaluation of the effects of the changing climate on PMP estimations, it suffers from a few limitations associated with the inconsistent spatial scales between precipitation data and models and the lack of adequate ground information for physical attributions. As shown before in the comparisons with previous studies, the discrepancies between statistical methods and hydrometeorological methods are evident (see Section 4.1 for details). They are mainly derived from the different rational behind the maximization framework, e.g., maximization of wind or moisture, and the uncertainty in the metrological data (e.g., dew point temperature, wind speed). The sensitivity of PMP estimation to different calculation methods is therefore worthy to be detected. However, we are unable to evaluate at this stage due to the lack of sufficiently long-term storm event records and related meteorological data for the maximization, especially on a national scale. Although we have validated our estimates with the auxiliary quasi-global PMP dataset and the in-situ observations, additional measures for quantitative validation, such as various methods, may further be employed in the future. Another limitation lies in the mathematical partition and subsequent attribution of the statistical PMP estimation into two main components following Eq. 8. Though the framework can conveniently be implemented for the attribution of PMP trends to different factors, no more sights of the dynamic and thermal atmospheric processes can be provided. Recent studies have shown the applications of numerical weather models (e.g., Weather Research and Forecasting





Model) in modeling the regional PMP (Hiraga et al., 2021). Such attempts are able to assess the sensitivity of PMP to different atmospheric (e.g., moisture) and geophysical factors (e.g., topography) and climate change from a physical perspective (Rastogi et al., 2017). Moreover, the scale difference of resolutions between the HRLT dataset (1 km) and GCM simulations (~1°, 100 km) may introduce regional disagreement between our historical assessment and future projections (Figure 7). This

difference is caused by the relatively coarse spatial resolution in the parameterization of GCMs, highlighting the fact that caution should be taken when explaining the linked spatial distribution between past and future.

Corresponding to the above-mentioned limitations, several strategies can be adopted to alleviate their impacts in future studies. A feasible solution is the use of multi-source meteorological data, for example, remote sensing product (e.g., MODIS-based vapor pressure data) and reanalysis predictions (e.g., ERA5 and JRA55), in the estimation of large-scale PMP using the

meteorological method, which can serve as a useful tool to verify the independent statistical estimations. Moreover, fully coupled regional-scale simulations can be performed using the numerical weather simulation and data assimilation techniques, of which WRF from NOAA has achieved much in the simulation and prediction of PMP (e.g., Rastogi et al., 2017). In addition to this, high-resolution global climate models such as the High-Resolution Model Intercomparison Project (HighResMIP v1.0) for CMIP6 provide another way for the PMP analysis on continental and global scales (Haarsma et al., 2016). However, inter-

member uncertainties that are inevitable and possibly considerable deserve more effort to constrain and alleviate. This point is highlighted by the comparison between historical CMIP PMP estimations and HRLT results (Figure 7) as well as the cross-comparison between CMIP and LFMIP-pdLC simulations in the past (Figure S6). The overestimated PMP and its components of the LFMIP-pdLC than the CMIP experiment during the baseline period can be a result of model sensitivity and uncertainty for the past climate. All of the issues discussed, including the unclear physical mechanisms of changing PMP and divergent

spatial scales among datasets and uncertainties therein, deserve to be studied in the future with the advancement of observation systems and earth system models.

## 5 Conclusions

Given the lack of knowledge in the spatial distribution of PMP in China and the potential influences of the changing climate on PMP formation, this study uses the existing most high-resolution (1 km) precipitation dataset to compute the 1d

PMP during 1961-2014 for the whole of China using the improved Hershfield method. The spatial distribution of PMP is generated on a national scale and has been validated with a satellite-based quasi-global PMP dataset and in-situ-based PMP results. Changes in PMP and its constituting factors ($X'_n$ and $K$) are presented in each 35-year time window from 1961-1995 to 1980-2014. Inter-annual trends are subsequently estimated during these periods and are attributed to the changes of these two contributors. An ensemble of GCMs is used to project the response of PMP to climate change under two scenarios (i.e.,

SSP126 and SSP585) in the near (2030-2064) and far future (2065-2099) of the 21st century relative to the baseline (1980-2014). The main findings are as follows:





(1) We find the approximately opposite spatial distribution of two constituting factors to form PMP ($X'_n$ and $K$) over the country, of which the variable $X'_n$ ($K$) generally decreases (increases) from the southeastern to the northwestern sections. They jointly result in a unique spatial distribution of PMP, which is featured by both the typical 'three steps' distribution from southeast to northwest and regional hotspots in coastal regions, mountainous areas, and northern arid zones. Our PMP estimations are generally consistent with previous precipitation compilations and project design results. However, overestimations are discovered when comparing with the in-situ-based PMP results and GPMM dataset, with correlation coefficients ranging from 0.65 to 0.96. The differences might be caused by the different calculation methodologies and varying spatial resolutions.

(2) Different temporal variations of $X'_n$ and $K$ are observed during moving time windows from 1961-2014. $K$ shifts from decrease to increase after the turning period of 1971-2005, while $X'_n$ keeps growing and achieved a 3% increase for the country. Consequently, PMP also increases from 106.5 to 109.5 mm/d from 1961-1996 to 1980-2014 period, with an accelerated speed after 1977-2011. The pattern suggests the increased flood control pressure in the context of simultaneously increasing reservoir capacity. The running trend of the 35-year PMP mainly lies in northern China, including inner Mongolia and Heilongjiang Provinces, which are predominately caused by the changes in the inter-annual variability (represented by the $K$ factor) together with the intensity of extreme precipitation (represented by $X'_n$). The PMP increases at a rate of 0.08 mm/day/a for the whole country, of which 71% (29%) is caused by the increasing $K$ factor ($X'_n$).

(3) The historical simulations of CMIP ensemble spatially agree with the HRLT results. Land-atmosphere coupling dominates the widespread increase in PMP over China under both SSP126 and the SSP585 climate change scenarios by modulating the intensity of daily precipitation extremes ($X'_n$), except for scattered regions in the Northwest China, where a significant increase in precipitation variability ($K$) is observed. No obvious differences in the future projections during the middle and end of the 21$^{st}$ century are discovered by comparing with the baseline. Nationally, the projected PMP changes are 17~20% and 0~2%, according to the SMIP and LFMIP-pdLC experiments under the SSP126 scenario, respectively. The percentages change to 43%~51% (SMIP) and -1~-5% (LFMIP-pdLC) for the SSP585 climate change scenario, indicating the strengthened modulations of land-atmosphere coupling to PMP with anthropogenic forcing.

Our study provides the first high-resolution map of PMP (1 d & 1 km) for China and quantitively challenges the reliability of the static climate assumption in conventional PMP estimation. Climate change and land-atmosphere coupling impacts are further projected using state-of-the-art ensemble models from CMIP6. Our results can provide scientific inferences to regional and national water managers and decision-makers for effective and efficient water resource management in the area.

**Author contributions**

Jinghua Xiong contributed to the data processing and wrote the original draft. Shenglian Guo and Abhishek contributed to the conceptual design and review of the manuscript. Shenglian Guo contributed to the funding acquisition and project administration. All co-authors reviewed and revised the manuscript.





**Competing interests**

The authors declare that they have no conflict of interest.

**Acknowledgments**

Jinghua Xiong thanks Dr. Qin Rongzhu and Prof. Zhang Feng from Lanzhou University for providing guidance on using the HRLT precipitation dataset.

**Financial support**

This work was supported by the National Key Research and Development Plan (grant number 2022YFC3202801) and Power China Huadong Engineering Corporation Limited (grant number 11SD210003A-01-2022).

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
