# Peer review of "Variation and attribution of probable maximum precipitation of"

_Hydrology and Earth System Sciences, 2023_

## Author Comment (AC1)

**Reply to Reviewers' comments (Reviewer#1)**

**Legend**
Reviewers' comments
Authors' responses
Direct quotes from the revised manuscript

**Reviewer #1:** The study emphasizes the importance of accurately assessing probable maximum precipitation (PMP) for high-risk water infrastructures, water resource management, and hydrological hazard mitigation. Using a fine spatiotemporal resolution precipitation dataset, the authors employ the improved Hershfield method to reveal the spatial distribution of 1-day PMP in China. They note a general decrease in national PMP from Southeast to Northwest. The research highlights the dominance of precipitation variability in North China and intensity in South China, with climate change expected to lead to a widespread increase in PMP (>20%) across the country under certain scenarios. Next, I offer comments indicating points where some clarifications could be helpful.

Response: We thank Dr. Simon Michael Papalexiou for his time in reviewing our manuscript and providing comprehensive suggestions for further improvements. Revisions for various sections have been made in the new version, as suggested. Please find our specific response to your comments below.

(1) Section 2.2: Note that it is not only the different products that can result in different estimates but also the resolution and methods used. Here, attempting to validate extrapolated results from the gridded HRLT resulted in bilinear interpolation adding extra uncertainty. The interpolation itself changes the values; known effects include smoothing out extremes (see e.g., 10.1175/JAMC-D-20-0259.1 for similar cases of re-gridding data). Additionally, the in-situ data are point measurements and can exhibit different behaviour than precipitation at 1km x 1km. The authors could comment on potential effects relevant with their methodological choices.

Response: Thank you very much for the notice. We fully agree with you that different choices of data sources, computation methods, and spatial scales can contribute to the discrepancies among different PMP estimations. We have added more explanations for the plausible influences on the validation of our PMP results in this section of the revised manuscript:
However, we note such spatial interpolation may introduce significant bias for the comparison of PMP estimations at different resolutions (Rajulapati et al., 2021), in addition to the systematic differences implicit to the methods and data sources (e.g., gauge vs satellite data).

(2) Section 2.3: There is a long literature regarding the evolution of the statistical method of PMP. I suggest the authors see the review of Salas et al. (2020; 10.1061/(ASCE)HE.1943-5584.0002003).

Response: Thank you for providing this comprehensive review. As suggested, we have enriched our introduction for the statistical algorithms applied in our study including the assumption, correction, and alterations of statistical PMP methods. The related content is copied below for easy review:
(1) The traditional statistical method was originally developed by Hershfield (Hershfield, 1961) based on Chow's frequency equation where a quantile of a probability distribution is expressed as a function of the mean, standard deviation, and a frequency factor $K_m$ (Chow, 1951). The frequency factor $K_m$ was estimated based on records of 24-h rainfall for 2700 stations in the United States (90% of the total stations) and subsequently modified to account for the effects of the sample size, outliers, and the difference between daily maximum and 24-h recorded data set (Hershfield, 1965, 1977). Salas et al. (2020) pointed out that the Hershfield method needs proper modification for applications in different climatic zones. Here, we employ an adjusted approach that has been widely applied for PMP design in China with the sampling bias and calendar day errors corrected (Lin, 1981; Hershfield, 1961).

(2) Apart from the traditional statistical methods to calculate PMP, many other methods have been proposed to describe the probabilistic nature of extreme precipitation events, though the

assumptions are shown to be unrealistic (Salas et al., 2014).

Reference:
Chow, V. T. (1951). A general formula for hydrologic frequency analysis. Trans. Am. Geophys. Union 32 (2): 231–237. https://doi.org/10.1029/TR032i002p00231.
Hershfield, D. M. (1961). Estimating the probable maximum precipitation. J. Hydraul. Eng. Div. ASCE. 87, 99-116.
    Hershfield, D. M. (1965). Method for estimating probable maximum rainfall. J. Am. Water Works Assn., 57(8): 965–972. https://doi.org/10.1002/j.1551-8833.1965.tb01486.x.
    Hershfield, D. M. (1977). Some tools for hydrometeorologists. In Proc., 2nd Conf. on Hydrometeorology. Boston: American Meteorological Society.
Lin, B. Z. (1981). Application of statistical estimation in study of probable maximum precipitation. Journal of Hohai University (Natural Sciences). 01, 52-63.
Salas, J. D., & J. Obeysekera. (2014). Revisiting the concepts of return period and risk under non-stationary conditions. J. Hydrol. Eng. 19 (3): 554–568. https://doi.org/10.1061/(ASCE)HE.1943-5584.0000820.
Salas, J. D., Anderson, M. L., Papalexiou, S. M., & Frances, F. (2020): PMP and Climate Variability and Change: A Review, J. Hydrol. Eng., 25, 03120002, https://doi.org/10.1061/(ASCE)HE.1943-5584.0002003.

(3) Section 2.5: Maybe I'm missing something here, but are the CMIP6 projections used raw? Daily projections of precipitation can have very large biases and probably cannot be used for this purpose. If these are bias-corrected projections, then what type of bias correction method was used? Since bias correction methods can also affect the results (see e.g., Luo et al. (2021; 10.1002/joc.7294, or Dong & Dong (2021; 10.1007/s00382-021-05773-1)

Response: We agree with the point that the original CMIP6 outputs can contain substantial bias in extreme precipitation that will inevitably impact the PMP projections. However, we could not perform the bias correction or the post-processing adjustments yet attempted to constrain the uncertainties in CMIP6 projections because the bias correction requires historical observations in the real world, which are impossible to acquire for the LFMIP-pdLC scenario where the land surface is prescribed by the long-term climatology. Given also the fact the chosen CMIP6 models (i.e., CMCC-ESM2, CNRM-CM6-1, EC-Earth3, IPSL-CM6A-LR, MIROC6, and MPI-ESM1-2-LR) reproduce fairly well observed daily extreme precipitation in China (Yang et al., 2021; Abdelmoaty et al., 2021). They are the only available models in the LFMIP experiments. We thereby choose the ensemble method to constrain uncertainties arising from model differences following previous studies (e.g., Qiao et al., 2023; Zhou et al., 2022). We additionally demonstrate the individual PMP projections for each model in the supplementary file to consider the outliers of the multi-model mean when the number of models is small. We have added some clarifications for these circumstances in the revised section as:

Despite the fact that the raw CMIP6 models can contain large bias for precipitation extremes, we could not perform the bias correction or the post-processing adjustments due to unavailable in-situ observations under the LFMIP scenarios. While the multi-model mean method is applied to constrain the individual model uncertainties in simulating precipitation extremes (e.g., Zhou et al., 2022; Qiao et al., 2023). The deviations across models are additionally illustrated in the supplementary files to reflect the model variance. Our findings provide a large-scale assessment of the future PMP changes over the country for policymaking and the local-scale investigations may further be supplemented by future field observations and climate models for informed decision-making.

Reference:
Abdelmoaty, H. M., Papalexiou, S. M., Rajulapati, C. R., & AghaKouchak, A. (2021). Biases beyond the mean in CMIP6 extreme precipitation: A global investigation. Earth's Future, 9(10), e2021EF002196. https://doi.org/10.1029/2021EF002196
Qiao, L., Zuo, Z., Zhang, R., Piao, S. L., Xiao, D., & Zhang, K. W. (2023). Soil moisture–atmosphere coupling accelerates global warming. Nat. Commun., 14, 4908. https://doi.org/10.1038/s41467-023-40641-y
Yang, X., Zhou, B., Xu, Y. et al. CMIP6 Evaluation and Projection of Temperature and Precipitation over China. Adv. Atmos. Sci. 38, 817–830 (2021). https://doi.org/10.1007/s00376-021-0351-4
Zhou, S., Williams, A. P., Lintner, B. R., Findell, K. L., Keenan, T. F., Zhang, Y., & Gentine, P. (2022). Diminishing seasonality of subtropical water availability in a warmer world dominated by soil moisture–atmosphere feedbacks. Nat. Commun., 13, 5756. https://doi.org/10.1038/s41467-022-33473-9

(4) Section 3.1: "...which is related to the systematic overestimation of GPM IMERG product over China." You can also check the review by Tang et al. (2020; 10.1016/j.rse.2020.111697) on the topic.

I am not sure if the linear regression presented in Figure 4a, b, c has meaning. It is clear from the figures that the linear regression line is not a good model. The nonlinearity is apparent from the cloud of points. Thus, conclusions and discussions based on the fitted linear regression can be misleading.

Response: Thank you for providing the useful reference that has been merged in our analysis of the revised manuscript. Moreover, we choose to retain our linear regression line in Figures 4a-4c for the evaluation, instead of the extrapolation, of our PMP estimations. Ideally, the point-scale PMP estimations from HRLT, GPM, and gauge records should well fit the 1:1 linear model in essence just like the basin-scale comparisons (Figures 4d-4f). We understand the cloud points of high precipitation may degrade the linear fitting and lead to a non-linear one. However, the relevant results remain uninfluenced and unchanged, i.e., underestimation of HRLT PMP compared to the in-situ and satellite estimates due to differences in data sources, calculation methods, and spatial resolutions.

(5) Section 3.3: Results shown here must be coupled with results from studies validating the performance of CMIP6 over China. As mentioned before, more details should be given on the projections used (raw, bias-corrected, what methods were used for BC, etc.). (See also Zhu et al. (2020; 10.1007/s00376-020-9289-1)

Response: As suggested, we have provided more explanations of CMIP6 model performances here and more details of data processing in the revised manuscript:
We find the ensemble mean PMP is the balanced result of overestimated values from CMCC-ESM2 and MIROC6 and the underestimated values from MPI-ESM1-2-LR, which is caused by the requisite interpolation from the native coarse model resolution (~2°). Historical evaluations have also shown the relatively better performance of EC-Earth and MPI-ESM1-2-LR than the remaining models with wet or dry biases over China (Dong and Dong, 2021; Jia et al., 2023). Despite the bias in raw CMIP6 outputs of different models, the multi-model ensemble has been shown as a useful method to reduce the uncertainty than individual models, which have the potential to further improve with future model evolution (Qiao et al., 2023; Zhu et al., 2020).

Reference:
Dong, T. Y., & Dong, W. J. (2021). Evaluation of extreme precipitation over Asia in CMIP6 models. Climate Dynamics, 57(7–8), 1751–1769. https://doi.org/10.1007/s00382-021-05773-1
Jia, Q., Jia, H., Li, Y., Yin, D. (2023). Applicability of CMIP5 and CMIP6 Models in China: Reproducibility of Historical Simulation and Uncertainty of Future Projection. Journal of Climate, 36 (17), 5809–5824, https://doi.org/10.1175/JCLI-D-22-0375.1
Qiao, L., Zuo, Z., Zhang, R., Piao, S. L., Xiao, D., & Zhang, K. W. (2023). Soil moisture–atmosphere coupling accelerates global warming. Nat. Commun., 14, 4908. https://doi.org/10.1038/s41467-023-40641-y
Zhu, H., Jiang, Z., Li, J., Li, W., Sun, C., & Li, L. (2020). Does CMIP6 inspire more confidence in simulating climate extremes over China? Advances in Atmospheric Sciences, 37(10), 1119–1132. https://doi.org/10.1007/s00376-020-9289-1

(6) Section 4.1: "Yang et al. (2018) discussed the potential of satellite data to estimate PMP in poorly gauged regions." Indeed, all these gridded products (and not only satellite-based) can offer an option when in-situ data are not available, but it must be clear that there are large differences in extreme precipitation values these products report (see Rajulapati et al. (2020; 10.1175/JHM-D-20-0040.1). The fact is that no single product can claim the best performance, and it seems the best option we have is to use many of them and quantify the uncertainty.

Response: Thank you for bringing the point to our attention. We have modified this statement and added this opinion to our discussion of the new version:
Using three remote sensing precipitation products and the statistical method, Yang et al. (2018) discussed the potential of gridded precipitation extremes to estimate PMP in poorly gauged regions by taking the Dadu River basin in western China (located in the upstream Yangtze River basin, Figure 1a) as an example. They pointed out the huge disparity among PMP values based on various satellite products (ranging between 51.88–519.11 mm, 90.16–417.61 mm, 122.41–391.79 mm, and 128.37–740.45 mm for CGDPA, CMORPH, PERSIANN-CDR, and TRMM 3B42V7, respectively) and recommended the PMP of 52~519 mm/d over the region, nearly two-fold higher than our result

of about 29~279 mm/d. The large differences among global precipitation products highlight the lack of consistent PMP representations in different areas, which may partly be solved by merging multiple data sources based on their regional performance and uncertainty quantification (Rajulapati et al., 2020).

Reference:
Rajulapati, C. R., Papalexiou, S. M., Clark, M. P., Razavi, S., Tang, G., & Pomeroy, J. W. (2020). Assessment of extremes in global precipitation products: How reliable are they? Journal of Hydrometeorology, 21(12), 2855–2873. https://doi.org/10.1175/JHM-D-20-0040.1
Yang, Y.; Tang, G.; Lei, X.; Hong, Y.; Yang, N. Can satellite precipitation products estimate probable maximum precipitation: A comparative investigation with gauge data in the Dadu River basin. Remote Sens. 2018, 10, 41. https://doi.org/10.3390/rs10010041

(7) Section 4.3: One additional point to mention could be the general limitations of PMP methods. What exactly do these methods offer compared to probabilistic methods? The literature on using probability methods in China is pretty rich (see, for example, a review by Gu et al. (2022; 10.1016/j.advwatres.2022.104144).

Response: We agree with the idea that distinguishing the meaning of various PMP calculation methods is important. We have added more discussions on this topic based on the current literature as:
Basically, these different methods to compute PMP have different storylines. For example, the hydro-meteorological methods are characterized by the maximization of a single or several atmospheric factors, and emphasize the physical mechanisms behind the storms (Gu et al., 2022). While the statistical methods estimate an unprecedented extreme value from a probabilistic perspective (Papalexiou and Koutsoyiannis, 2006; Papalexiou et al., 2016). The hydro-meteorological methods may be somewhat more physically realistic than statistical methods but they heavily rely on the meteorological data and neglect the interaction of different factors, which are unable to estimate the large-scale and future projection PMP accurately.

Reference:
Gu, X., Ye, L., Xin, Q., Zhang, C., Zeng, F., Nerantzaki, S.D., Papalexiou, S.M. (2022). Extreme precipitation in China: a review. Adv. Water Resour., 163, p. 104144, 10.1016/j.advwatres.2022.104144.
Papalexiou, S. M. & Koutsoyiannis, D. (2006). A probabilistic approach to the concept of probable maximum precipitation, Adv. Geosci., 7, 51–54, https://doi.org/10.5194/adgeo-7-51-2006.
Papalexiou, S. M., Dialynas, Y. G., & Grimaldi, S. (2016). Hershfield factor revisited: Correcting annual maximum precipitation, J. Hydrol., 542, 884–895, https://doi.org/10.1016/j.jhydrol.2016.09.058.

Overall, I believe this is a well-written and well-organized study that has its place in the literature, pending some minor amendments to clarify some points. I enjoyed reading the paper, and if PMP methods are still officially used in engineering practice in China, then this could be a useful study for practitioners. Most of my discussion points are optional, and the authors can skip them. The most important ones are related to clarifications about the CMIP6 outputs used and whether they have been biased corrected, etc.

Response: We thank Dr. Simon Michael Papalexiou again for his time in reviewing our manuscript and the useful suggestions on it. We hope the comments have been properly addressed by additional discussions and clarifications, which have improved the manuscript. Since the PMP methods is recommended by WMO and still serve as the check flood for important water conservancy and hydropower projects in China, our manuscript will potentially be of good value for hydraulic practitioners and the climate community, as you rightly indicated.

---

## Author Comment (AC2)

**Reply to Reviewers' comments (Reviewer#2)**

**Legend**
Reviewers' comments
Authors' responses
Direct quotes from the revised manuscript

**Reviewer #2:** This study employs a high-resolution (1d, 1km) precipitation dataset to estimate spatially distributed PMP across China, and compares these estimates with two benchmark datasets. It focuses on analysing the trend of PMP over time, rather than providing a static estimate, and extends its investigation into future changes in PMP based on climate model outputs. The discussion section is thorough, touching upon several interesting topics. Overall, the manuscript is well-written, offering insightful findings and valuable data pertinent to China. I have no significant concerns but would like to offer a few suggestions for consideration in the manuscript revision:

Response: We thank Dr. Tang Guoqiang for his time in reviewing our manuscript and providing useful suggestions for improvement. Revisions for the manuscripts have been made in the new version, as suggested. Please find our specific response to your comments below.

**Specific comments:**
(1) Line 15: The manuscript mentions that the dataset integrates observations with machine learning algorithms (Section 2.1). Including a brief explanation (e.g., a few words) here could offer valuable context to readers.

Response: As suggested, we have added descriptions for the precipitation dataset in the Abstract to make it more informative to readers as:
Here, we use the finest spatiotemporal resolution (1d & 1km) precipitation dataset from an ensemble of machine learning algorithms to present the spatial distribution of 1d PMP based on the improved Hershfield method.

(2) Paragraph Structure: The manuscript frequently utilizes long paragraphs. I suggest breaking these into shorter, more digestible sections to enhance readability.

Response: Thanks for the suggestion. We have modified the paragraph structures throughout the revised manuscript for better readability.

(3) Figure 3: To aid in comparison, I recommend employing a consistent colour scale across all sub-figures.

Response: We have revised the figure into a consistent colour scheme for better inter-comparison in the new version (Figure 3):

[Figure]

**Figure 3: (a) Spatial distribution of field-based PMP over 80 secondary river basins (Wang, 2002). (b) Spatial distribution of recorded historical maximum daily precipitation (Wang, 2002, Table S4). (c) Spatial distribution of PMP based on in-situ daily precipitation during 1961-2014. (d) Spatial distribution of PMP results from the GPMM database.**

(4) Figure 6: The patchy patterns observed, particularly in panels (d) and (e), warrant further explanation. Clarifying these patterns could enhance the reader's understanding of the underlying

data and analysis.

Response: The patchy patterns are caused by the opposite direction of trends in $X'_n$ and $K$, leading to completely different distributions. This is more obvious where the PMP trend is dominated by $K$, which generally offsets the influences of $X'_n$ and therefore presents extremely high and locally variable contributions. We have added more explanations for this spatial distribution in the revised manuscript as:

We observe the opposite trends in $X'_n$ and $K$ nationwide, resulting in the patterns of extremely high (low) relative contribution of $K$ ($X'_n$) over regions where PMP changes are controlled by the former (Figures 6d and 6e).

(5) Figure 8: The substantial discrepancy between UserSMIP and LFMIP-pdLC raises questions about the reliability of the findings. Incorporating the other reviewer's (Simon) comment for CMIP6 validation and bias correction might address these concerns and strengthen the manuscript.

Response: Thank you very much for the comment. The large differences between the two experiments are considered influences of land-atmosphere coupling, i.e., from the prescription of the climatology of soil moisture and snow. While we acknowledge considerable bias may exist in GCM outputs, the bias correction could not be performed because our study emphasizes the pair comparisons between ideal experiments, while the real-world observations are not available under such scenarios (i.e., LFMIP-pdLC). Given also the fact the chosen CMIP6 models (i.e., CMCC-ESM2, CNRM-CM6-1, EC-Earth3, IPSL-CM6A-LR, MIROC6, and MPI-ESM1-2-LR) reproduce fairly well observed daily extreme precipitation in China (Yang et al., 2021; Abdelmoaty et al., 2021). We attempt to constrain such uncertainties using the ensemble method with more available models (e.g., Qiao et al., 2023; Wei et al., 2023). We apply a total of six models that are the only ones providing daily precipitation variables in both experiments. We demonstrate the ensemble mean and individual projection of each model in the supplementary files to show the potential uncertainty range. Such procedures, to an extent, provide a reliable large-scale PMP projection over the country, which is the focus of this study.

We have added more explanations and justifications in Section 2.5 of the revised manuscript:

Despite the fact that the raw CMIP6 models can contain large bias for precipitation extremes, we could not perform the bias correction or the post-processing adjustments due to unavailable in-situ observations under the LFMIP scenarios. While the multi-model mean method is applied to constrain the individual model uncertainties in simulating precipitation extremes (e.g., Zhou et al., 2022; Qiao et al., 2023). The deviations across models are additionally illustrated in the supplementary files to reflect the model variance. Our findings provide a large-scale assessment of the future PMP changes over the country for policymaking and the local-scale investigations may further be supplemented by future field observations and climate models for informed decision-making.

Reference:
Qiao, L., Zuo, Z., Zhang, R., Piao, S. L., Xiao, D., & Zhang, K. W. (2023). Soil moisture–atmosphere coupling accelerates global warming. Nat. Commun., 14, 4908. https://doi.org/10.1038/s41467-023-40641-y
Wei, L., Xin, X., Li, Q., Wu, Y., Tang, H., Li, Y. & Yang, B. (2022). Simulation and projection of climate extremes in China by multiple coupled model Intercomparison project Phase 6 models. Int. J. Climatol. 43, 219–239. https://doi.org/10.1002/joc.7751
Abdelmoaty, H. M., Papalexiou, S. M., Rajulapati, C. R., & AghaKouchak, A. (2021). Biases beyond the mean in CMIP6 extreme precipitation: A global investigation. Earth's Future, 9(10), e2021EF002196. https://doi.org/10.1029/2021EF002196
Yang, X., Zhou, B., Xu, Y. et al. CMIP6 Evaluation and Projection of Temperature and Precipitation over China. Adv. Atmos. Sci. 38, 817–830 (2021). https://doi.org/10.1007/s00376-021-0351-4